# Senescent Markers Expressed by Periodontal Ligament-Derived Stem Cells (PDLSCs) Harvested from Patients with Periodontitis Can Be Rejuvenated by RG108

**DOI:** 10.3390/biomedicines11092535

**Published:** 2023-09-14

**Authors:** Ilaria Roato, Giacomo Baima, Clarissa Orrico, Alessandro Mosca Balma, Daniela Alotto, Federica Romano, Riccardo Ferracini, Mario Aimetti, Federico Mussano

**Affiliations:** 1Department of Surgical Sciences, University of Turin, 10126 Turin, Italy; ilaria.roato@unito.it (I.R.); alessandro.moscabalma@unito.it (A.M.B.); federica.romano@unito.it (F.R.); mario.aimetti@unito.it (M.A.); federico.mussano@unito.it (F.M.); 2Politecnico di Torino, 10129 Turin, Italy; 3Fondazione Ricerca Molinette-Onlus, A.O.U. Città della Salute e della Scienza, 10129 Turin, Italy; issa.93@live.it; 4Skin Bank, Department of General and Specialized Surgery, A.O.U. Città della Salute e Della Scienza, 10126 Turin, Italy; daniela.alotto@gmail.com; 5Department of Surgical Sciences and Integrated Diagnostics, University of Genoa, 16132 Genoa, Italy; riccardoferraciniweb@gmail.com

**Keywords:** cell therapy, mesenchymal stem cells, periodontal diseases, senescence, periodontal regeneration, rejuvenation

## Abstract

Periodontal ligament (PDL) has become an elective source of mesenchymal stem cells (PDLSCs) in dentistry. This research aimed to compare healthy PDLSCs (hPDLSCs) and periodontitis PDLSCs (pPDLSCs) to ascertain any possible functional differences owing to their milieux of origin. Cells were tested in terms of colony-forming unit efficiency; multi differentiating capacity; immunophenotype, stemness, and senescent state were studied by flow cytometry, immunofluorescence, and β-galactosidase staining; gene expression using RT-PCR. Both hPDLSCs and pPDLSCs were comparable in terms of their immunophenotype and multilineage differentiation capabilities, but pPDLSCs showed a senescent phenotype more frequently. Thus, a selective small molecule inhibitor of DNA methyltransferase (DNMT), RG108, known for its effect on senescence, was used to possibly reverse this phenotype. RG108 did not affect the proliferation and apoptosis of PDLSCs, and it showed little effect on hPDLSCs, while a significant reduction of both p16 and p21 was detected along with an increase of SOX2 and OCT4 in pPDLSCs after treatment at 100 μM RG108. Moreover, the subset of PDLSCs co-expressing OCT4 and p21 decreased, and adipogenic potential increased in pPDLSCs after treatment. pPDLSCs displayed a senescent phenotype that could be reversed, opening new perspectives for the treatment of periodontitis.

## 1. Introduction

Periodontal ligament (PDL) was proposed as a source of cells capable of repairing multiple tissues in the ‘70s of the past century [1]. In 1985, McCulloch identified progenitor cells around the blood vessels of the PDL [2], but it was only in 2004 that Seo and coworkers isolated and characterized human PDL stem cells (PDLSCs) [3]. PDLSCs were similar to other MSCs of the oral cavity in terms of differentiation potential (osteogenic, adipogenic, chondrogenic, and neurogenic) and colony-forming units (CFU) capabilities [4]. They could also deposit cement-like structures endowed with condensed collagen fibers resembling PDL in animal models, representing a promising source for tissue engineering [5]. Nonetheless, the use of PDLSCs may be hindered by their scarcity or complete unavailability, since the patient has to present, for example, impacted third molars or bicuspid teeth to be extracted for orthodontic reasons [6].

In addition, due to the high global prevalence of periodontitis [7], a matter of growing interest is the role played by a chronic inflammatory environment on stem cells as regards their potency and regenerative capabilities [8]. Inflamed PDL has proven to be a valuable source of stem cells, which may also show a neurodifferentiation capability [9,10]. Park et al. confirmed the feasibility of harvesting PDLSCs from inflamed human PDLs and demonstrated that these cells called inflammatory PDLSCs (iPDLSCs), retained their regenerative potential [11]. Liu et al. pointed out, however, that iPDLSCs display impaired immunomodulatory properties [12]. This outcome was also confirmed in a further study that compared iPDLSCs to healthy hPDLSCs, describing for the former an increased proliferation rate, higher migratory capacities, decreased osteogenic potential, and lesser quality in the cell sheets obtained [13]. This data appeared relevant considering that iPDLSCs and hPDLSCs were obtained, respectively, from a periodontally compromised and a healthy tooth of the same patient to avoid donor variation.

In the last years, researchers have focused on ascertaining how the inflammatory micro-environment alters stem cells in the PDL and whether any strategies may be envisaged to reduce the functional differences between iPDLSCs and hPDLSCs. To explain this phenomenon, a replicative senescence may be invoked, as it is a stress-dependent mechanism, in which telomere shortening triggers the p53-dependent pathway resulting in p21 activation [14]. Senescent cells also express p16Ink4a which may contribute to cell growth arrest and stem cell exhaustion [15]. Although mitosis is the main event causing telomere shortening, a series of stressors, such as oxidative stress [16], cigarette smoke [17], and chronic exposure to bacterial virulence factors such as cytolethal distending toxins (CDTs) [18] may engender replicative senescence, that is characterized by beta galactosidase positive staining and apoptosis resistance. This senescent-like phenotype, independent of telomere shortening, is called stress-induced premature senescence [19]. Replicative senescence, however, may reinforce senescence phenotype even by partially accelerating the telomere shortening [20]. 

The expression of telomerase reverse transcriptase (TERT), and thus a modulation of telomere length, can be induced by blocking methylation at the TERT promoter region through DNA methyltransferase inhibitor (DNMTi) drugs like RG108, which could achieve positive effects on cellular senescence, potentially expanding the application of stem cell therapy [21]. Recently, RG108 has been tested on bone marrow derived-MSCs of swine-origin obtaining an effect on senescence, apoptosis, and pluripotency gene expression [22]. In light of these premises, this research is aimed at comparing hPDLSCs and periodontitis patient-derived PDLSCs (pPDLSCs), according to their single or double expression of stemness and senescent markers. Moreover, RG-108 was applied to test its rejuvenating effect on both cell types. 

## 2. Materials and Methods

### 2.1. Isolation and In Vitro Culture of PDLSCs

PDLSCs were isolated from wisdom teeth of 5 healthy donors and of 12 periodontitis patients (pPDLSCs), in accordance with ethical principles. The study was approved by the local Ethics Committee (n° 0107683). Briefly, the dental root was scratched, and the small fragments of tissue were washed with physiological saline buffer (PBS), then digested for 10 min with a solution of 1 mg/mL dispase II (PluriSTEM™, Merk) and 3 mg/mL collagenase I (Sigma Aldrich, Burlington, MA, USA). Afterwards the digested fragments of tissue were washed and seeded in a petri dish (10 cm diameter) in α-MEM with 15% foetal bovine serum (FBS), 5% penicilin/streptomicin, 1% amphotericin B (Life Technologies, Carlsbad, CA, USA). The culture dishes were then incubated at 37 °C in a humidified atmosphere containing 5% CO_2_. After 7 days, the medium was refreshed, leading the cells to grow until confluence for 15–20 days. Cells were then detached, counted, and expanded in a medium promoting the growth of mesenchymal cells, containing α-MEM + Glutamax, with 5% of human platelet lysate (hpl), (IsoCell Growth, Euroclone, Pero, Italy), and 5% penicilin/streptomicin (Life Technologies, Carlsbad, CA, USA).

### 2.2. Colony-Forming Unit (CFU) Assay

Single-cell suspensions of hPDLSC and pPDLSCs were plated in 6-multiwell plates (Corning, Lowell, MA, USA) in a basal medium at a density of 0.6 × 10^3^ cells/well and were cultured for 2 weeks (the medium was refreshed every 3 days). The cells were then fixed with 4% paraformaldehyde and stained with crystal violet. Colonies were visualized under the microscope and pictures were taken.

### 2.3. RG108 Preparation Population Doubling Time Assay (PDT)

RG108 was purchased from Miltenyi Biotech (Bergisch Gladbach, Germany), reconstituted in DMSO (Sigma Aldrich, Burlington, MA, USA) according to the instructions, and stored at −20 °C. To determine the population doubling time (PDT), both hPDLSCs and pPDLSCs were seeded in T25 flasks at three different densities with standard medium alone, with 50 μM, 100 μM RG108, or dimethyl sulfoxide (DMSO), this last condition to exclude toxicity due to the vehicle of RG108. The experiment was repeated 5 times for both hPDLSCs and pPDLSCs. Medium and RG108 were replaced after 3 days, then cells were detached and counted after 5 days. To assess the population doubling time (PDT), the log natural l(n) for the initial concentrations was computed. Afterwards, the angular linear coefficient (μ) was obtained from a l(n) linear regression (l(n)–dependent variable x time–independent variable). PDT was computed using the following equation g = ln2/μ (ln2 = 0.6931).

### 2.4. Multi-Differentiating Cell Culture Conditions

To study PDLSC multi-differentiating capabilities, the cells were seeded in different media. Osteogenic medium (OM) was composed of α-MEM supplemented with 10% FBS, 50 μg/mL ascorbic acid, 10^−8^ M dexamethasone, and 10 mM beta-glycerophosphate (Sigma Aldrich, Burlington, MA, USA) and cultures were maintained for 30 days. Then, the formation of a mineralized matrix was assessed by Von Kossa staining (Sigma Aldrich, Burlington, MA, USA). Adipogenic and chondrogenic media were purchased from Miltenyi (Miltenyi Biotech, Bergisch Gladbach, Germany) and cultures were maintained for 21 days. The formation of adipocytes were detected by oil red O staining, which colors lipid droplets. Chondrogenic differentiation was detected by Aggrecan (ACAN) staining of chondrocyte micromasses.

### 2.5. RNA Isolation and Real-Time

RNA was isolated according to the TRIzol procedure (Thermo Fisher Scientific, Carlsbad, CA, USA), 1 µg of RNA was retrotranscribed to single-stranded cDNA by the High-Capacity cDNA Reverse Transcription Kit from Applied Biosystems (Thermo Fisher Scientific, Carlsbad, CA, USA).

The mRNA expression of the following genes was tested: NANOG, SOX2, OCT4, RUNX2, SOX9, PPARg, p21, p16, BCL2. The primer sequences were synthesized by Life Technologies and reported in Table 1. RT-PCR was performed with Luna^®^ Universal qPCR Master Mix (New England BioLabs, Ipswich, MA, USA), using the CFX96 system (Bio-Rad, Hercules, CA, USA). The amplification protocol foresees 40 cycles with an annealing temperature of 58 °C or 60 °C, according to the primers. The expression of β-actin was chosen to normalize gene expression data and the 2−ΔΔCt method was used for the quantitative analysis using CFX Manager software (Bio-Rad, Hercules, CA, USA). Gene expression data are presented as mean ± SEM.

### 2.6. Flow Cytometry

The expression of typical MSCs markers was analyzed by flow cytometry on freshly isolated PDLSCs through MSC phenotyping kit (Miltenyi Biotech, Bergisch Gladbach, Germany), while cells expanded and maintained in cultures were then identified as CD105, CD73, CD90, CD44 positive and CD45 negative. Cells grown in culture were detached and a standard labelling protocol for surface antigens was performed with the following monoclonal antibodies fluorochrome-conjugated and isotypic controls: human CD105 PE (Invitrogen, Camarillo, CA, USA), CD73 FITC, CD44 FITC, CD45 PerCP, IgG1 PE, IgG1 FITC and IgG2a PerCP (Miltenyi Biotech, Bergisch Gladbach, Germany), CD90 PerCP (Biolegend, San Diego, CA, USA). As a control, unstained cells were examined. For detecting OCT4 and p21, an intra-nuclear and cytoplasmatic staining was adopted, according to the Miltenyi procedure of the Transcription Factor Staining Buffer Set (Miltenyi Biotech, Bergisch Gladbach, Germany). For both the molecules monoclonal antibodies OCT4 by Miltenyi Biotech (Bergisch Gladbach, Germany) and p21 by Thermo Fischer Scientific (Carlsbad, CA, USA) were used. A dye for nucleic acid 7-Amino-Actinomycin D (7-AAD) and phospholipid-binding protein Annexin V (BD Pharmingen, San Jose, CA, USA) were used as death markers, as per the manufacturer’s instruction. Data were obtained on a MACsQuant 10 and computed with MACsQuantify software (Miltenyi Biotech, Bergisch Gladbach, Germany). The data is presented as percentages of cells expressing specific markers (mean ± SD).

### 2.7. β-Galactosidase Staining

Both hPDLSCs and pPDLSCs were cultured for 2 passages and the rate of senescence was evaluated through a β-Galactosidase staining according to the manufacturer’s instruction of the senescence β-Galactosidase staining kit (Cell Signaling Technology, Danvers, MA, USA). Briefly, PDLSCs were plated in 6-multiwell plates for 24 h, then the medium was removed, cells washed, fixed, and stained with the β-Galactosidase solution overnight in a dry incubator. The blue β-Galactosidase staining was visualized under a microscope.

### 2.8. Immunofluorescence

Immunofluorescent staining was employed to assess the presence of OCT4 and p21. Initially, PDLSCs were cultured in 8-multiwell chambers from Thermo Fisher Scientific (Carlsbad, CA, USA). Following a 24-h incubation at 37 °C to facilitate cellular adhesion, the cells were fixed using 4% paraformaldehyde, permeabilized using TBS containing 0.5% Triton, and then blocked with TBS containing 3% bovine serum albumin (BSA; Sigma-Aldrich). The subsequent steps involved incubating the cells with primary antibodies; p21 was incubated for 1 h at room temperature, while OCT4 required 3 h. This was followed by an incubation with secondary antibodies conjugated to a fluorophore for 1 h at room temperature, as reported elsewhere [23] Unconjugated monoclonal antibodies directed to human OCT4 (9B7), p21 (R.229.6) and their respective secondary antibodies goat anti-Mouse IgG1 Alexa Fluor 568 and goat anti-rabbit IgG DyLight 488 were purchased from Thermo Fisher Scientific (Carlsbad, CA, USA). Nuclei were stained through SlowFade Gold antifade reagent with DAPI by Invitrogen (Thermo Fisher Scientific). Analysis was performed through a Leica SP8 inverted confocal microscope (Leica, Wetzlar, Germany).

### 2.9. Statistical Analyses

Prism software (version 9.0) was utilized for statistical analysis. Data were subjected to Student’s *t*-test to evaluate statistically significant differences between hPDLSC and pPDLSC groups. A one-way ANOVA using the Bonferroni post hoc test was performed to analyze gene expression data among the different groups (CTRL, DMSO, and RG108 treated). Statistical significance was considered with *p* < 0.05.

## 3. Results

### 3.1. Immunophenotype of hPDLSCs and pPDLSCs Is Comparable

PDL tissues derived from compromised teeth in periodontitis patients and healthy donors were enzymatically digested, obtaining a heterogeneous population of cells (Figure 1A), which contains a small percentage of CD105/CD73/CD90+ CD45- MSCs cells (Figure 1B) along with non-MSCs (CD14, CD19, CD34, CD45+ cells) and for epithelial cells (EpCam1+ cells), data was not shown. Both hPDLSCs and pPDLSCs, put in culture in the mesenchymal growth medium, gave origin to small colonies, as shown by crystal violet staining (Figure 1C), then they grew until confluence (p0) and subsequently expanded for a further 8, in vitro passages. The immunophenotype of PDLSCs from p0 to p3 was analyzed and compared, showing a comparable percentage of MSCs in patients and healthy donors during the passages, except at p3, where it was higher in pPDLSCs, *p* < 0.05 (Figure 1D). In the following passages, the percentage of MSCs remained high and comparable between the two groups.

### 3.2. hPDLSCs and pPDLSCs Show Multilineage Differentiation Capabilities

The expression of master genes for chondrocytes, adipocytes, and osteoblasts was analyzed in both hPDLSCs and pPDLSCs, showing comparable levels of expression for SOX9 and RUNX2 (Figure 2A,C), while the level of PPARg expression was higher in pPDLSCs than in hPDLSCs, *p* < 0.05 (Figure 2B). Then, both hPDLSC and pPDLSCs were cultured in the standard mesenchymal growth medium and specific differentiating medium. The chondro-differentiation medium induced the formation of chondromasses containing ACAN+ chondrocytes, (Figure 2E–G), while the standard medium chondromasses did not show chondrocytes (Figure 2D–F). Adipo-differentiation medium induced Oil red O adipocytes (Figure 2I–M), which were not present in the standard medium (Figure 2H–L). The osteogenic medium induced osteoblasts forming mineralized nodules (Figure 2O–Q), which were not detectable in a standard medium (Figure 2N–P).

### 3.3. pPDLSCs Show a Senescent Phenotype

One of the features of MSCs is the expression of the stemness genes, thus the level of expression of SOX2, OCT4, and NANOG was evaluated, showing that they were more expressed in pPDLSCs than in hPDLSC (Figure 3A). Since, we observed that by culturing cells for more passages, pPDLSCs had a trend to grow slowly and assumed a square shape, losing the typical spindle shape compared to hPDLSCs, we studied the expression of two senescence markers, p16 and p21, showing that their level is significantly higher in pPDLSCs than in hPDLSCs (Figure 3B). Thus, we hypothesized that pPDLSCs could undergo a sort of replicative senescence, and we stained them for β-Gal at an early passage. Among hPDLSCs the β-Gal staining was negative, while for pPDLSCs many cells were positive, confirming the senescent state (Figure 3C,D). Moreover, through flow cytometry, we showed that a subset of PDLSCs can contemporarily express stemness and senescent markers, OCT4 and p21 respectively, and this subset was higher in pPDLSCs than in hPDLSCs (38 ± 16.3 vs. 14.7 ± 10.9, mean ± SD) (Figure 3E). The immunofluorescence staining confirmed the expression of OCT4 on both hPDLSCs and pPDLSCs, but in pPDLSCs, more cells co-expressed OCT4 and p21 (Figure 3F,G).

### 3.4. Effect of RG108 on Proliferation and Apoptosis of PDLSCs

We tested two RG108 concentrations (50 μM, 100 μM) for their effects on PDT and cell death. After 5 days of culture, we did not observe any significant variation in the growth rate according to the different culture conditions for hPDLSCs and pPDLSCs (Figure 4A). None of the RG108 concentrations significantly affected PDT, compared to the non-treated PDLSCs nor for hPDLSCs and pPDLSCs (Figure 4B). The expression analysis of the anti-apoptotic gene BCL2 showed that pPDLSCs expressed it significantly less compared to hPDLSCs (Figure 4C). The treatment with 100 μM RG108 caused an increase in BCL2 expression in pPDLSCs similar to the one in hPDLSCs (Figure 4C). Looking at the apoptotic marker, Annexin V was more expressed in pPDLSCs compared to hPDLSCs, even though the variation did not reach statistical significance (Figure 4D–F).

### 3.5. RG108 Affected the Senescent Phenotype

We evaluated the effect of RG108 treatment on p16 and p21 expression, showing no significant variations for hPDLSCs, while a significant reduction of both genes was detected with the dose of 100 μM for pPDLSCs (Figure 5A,B). We also evaluated whether RG108 treatment could also affect the expression of stemness genes, showing no significant variations in hPDLSCs (Figure 5C) and an increase of SOX2 and OCT4 in pPDLSCs after treatment at 100 μM RG108 (Figure 5D). Then, we checked whether the subset of PDLSCs co-expressing OCT4 and p21 was modified by RG108, showing their decrease in pPDLSCs at 100 μM RG108, while no significant modulation was detected on hPDLSCs (Figure 5E).

### 3.6. RG108 Stimulates the Adipogenic Potential in pPDLSCs

We investigated whether RG108 determines the modulation of the multi-differentiating capabilities in PDLSCs. For both hPDLSCs and pPDLSCs the chondrogenic and osteogenic capabilities were not affected (Figure 6A–D). The expression of the master gene PPARγ was really low in hPDLSCs and it was not modulated by RG108, while a significant up-regulation was detected in pPDLSCs (Figure 6E,F) and culture in adipogenic condition resulted in the major formation of cells containing lipid droplets in pPDLSCs treated with 100 μM RG108 (Figure 6G–I).

## 4. Discussion

Given the growing importance of PDLSCs for tissue regeneration applications, this research aimed at comparing hPDLSCs and pPDLSCs, also assessing the possible role of RG-108 in reversing some deleterious effects that the inflamed environment exerts on pPDLSCs. To achieve a thorough characterization of PDLSCs, the authors presented the immunophenotypic characterization of the cells as soon as they were derived from the mechanic harvest followed by the enzymatic digestion of PDL (prior to p0). This was indeed a heterogeneous cell population containing only a small percentage of MSCs according to the phenotype panel established by the International Society for Cellular Therapy [24], along with a component of non-MSCs and epithelial cells, which are usually lost during the in vitro passages owing to the cell media used, which promote MSCs. By further monitoring the immuno-phenotype for three passages, a statistically significant difference (*p* < 0.05) emerged at p3 in terms of a higher presence of MSCs’ markers in the cells derived from compromised teeth in periodontitis patients (Figure 1D). Interestingly, CD73 and CD90 were always higher than 95%, but CD105 was overall less expressed and more susceptible to the cell milieu (healthy vs. inflamed). Recently, the role of CD105 in MSCs has gained attention owing to its possible association with immunomodulatory functions [25], besides the differences depending on cell culture conditions and passages [26]. CD105-MSCs have a significant inhibitory effect on the expression of lymphocytes in comparison with CD105+ cells [27]. Here, we showed that pPDLSCs expressed significantly more CD105 than hPDLSCs at p3, which is consistent with the decrease of MSCs immunosuppressive properties illustrated by Klinker et al. along subsequent passages [28].

To unveil any functional differences between hPDLSCs and pPDLSCs, their multi-differentiation potential toward cartilage, adipose tissue, and bone was assessed based on the mRNA expression of their respective master genes and tissue production in vitro. While SOX9 and RUNX2 were similarly expressed, which was in accordance with the formation of ACAN+ chondrocytes and mineralized nodules, the level of Peroxisome Proliferator-Activated Receptor-γ (PPARγ) was significantly higher in pPDLSCs than in hPDLSCs, (*p* < 0.05), which was, at least in part, portrayed by the formation of lipidic droplets visualized through the Oil red O staining. PPARγ is a ligand-activated transcription factor that binds PPAR response elements within the nucleus to modulate the expression of several genes involved not only in cell differentiation and metabolism but also in inflammation [29]. PPARγ is highly expressed in several cells including endothelial cells, smooth muscle cells, and monocyte/macrophages and it was involved in mitigating the effects of inflammation in a series of studies based on PPARγ agonists [30]. The increased expression of PPARγ in pPDLSCs reported is unprecedented and consistent with a possible defensive response to the persistent inflammation of the environment from which they were harvested.

Periodontitis is an inflammatory condition caused by a gradual change from symbiotic to pathogenic oral microbiota at the subgingival level [31]. Toxins released by periodontal pathogens, such as *Aggregatibacter actinomycetemcomitans* and *Porphyromonas gingivalis* showed genotoxic effects [18]. *P. gingivalis* releases lipopolysaccharide (LPS) and gingipains [32], causing continuous cellular stress, which contributes to the replicative senescence condition of PDLSCs and dental pulp cells [33]. Also, LPS has been shown to be able to promote senescent osteocyte accumulation in young alveolar bone [34], leading eventually to tooth loss. Indeed, persistent stimulation by LPS can cause genomic and oxidative damage resulting in a cellular senescent phenotype [35]. In addition, periodontitis promotes a chronic low-grade inflammatory state, characterized by elevated concentrations of cytokines, acute-phase proteins, and dysregulated metabolic markers [36,37,38]. We hypothesize that the chronic inflammatory conditions elicited by periodontitis both locally and systemically eventually lead to the increase of a subset of PDLSCs expressing both stemness and senescent genes, in an attempt to face the bacterial challenge and regenerate damaged PDL. Indeed, in physiological conditions senescent cells can be considered as a trigger of tissue remodeling, which is a multi-step process; whereas senescent cells eliminate themselves, since they are damaged cells, to promote tissue renewal [39]. In pathological conditions, such as periodontitis, this process starts but it is unsuccessful since senescent cells accumulate exacerbating chronic inflammation without tissue regeneration [40,41]. Moreover, the initial release of factors, constituting the senescence-associated secretory phenotype (SASP) [42] normally should promote plasticity and tissue regeneration, but prolonged exposure to the SASP causes an accumulation of senescent cells [43]. Our data follow this altered regenerative process, indeed we reported an increased expression of stemness genes in pPDLSCs compared to hPDLSCs, but these cells up-regulate also senescent markers, such as p16, p21, and β-gal, leading to a phenotype more similar to the replicative senescent one. It has been particularly interesting to find a heterogeneous population of PDLSCs according to the different expression of stemness and senescent markers because it is known that p21 and p16 increase in aging PDLSCs [44,45], but the contemporary expression of p21 and OCT4 in a subset of PDLSCs is unprecedented to the best of our knowledge. We also evaluated whether pPDLSCs have alterations of molecules involved in the control of the apoptotic process and we detected a significantly lower expression of BCL2 in pPDLSCs than in hPDLSCs. Conversely, Annexin V was slightly more expressed in pPDLSCs than in hPDLSCs. BCL-2 is a negative regulator of apoptosis [46] and we hypothesized that a reduced BCL2 expression by pPDLSCs can be due to the stress conditions in the periodontitis microenvironment, enriched with free oxygen radicals, indeed it has been demonstrated that PDLSCs treated with H_2_O_2_ showed inhibition of BCL2 [47]. On the other side, the reduced BLC2 expression by pPDLSCs suggests that these cells are not again arrived in a senescent replicative state [48], thus they could be rescued.

PDLSCs show multi-differentiating potential, indeed, in our series hPDLSCs and pPDLSCs showed comparable levels of the osteogenic and chondrogenic master genes, while the expression of the adipogenic PPARγ was higher in hPDLSCs than in pPDLSCs. PPARγ is known to be affected by a series of different stimuli and conditions in PDLSCs subjected to an inflammatory environment. It is the case, for instance, of LPS, which upregulated PPARγ mRNA expression along with SOX9 [49]. Furthermore, high levels of glucose, such as in diabetic patients, promote the expression of PPARγ, which is detrimental to the reparative potential of the periodontal defect [50]. Since changes in cell phenotype during osteoblastogenesis, chondrogenesis, and adipogenesis induce suppression of stemness genes and concurrent activation of tissue-specific-related genes, we focused our attention on molecules that are potentially able to control these gene regulations. In particular, epigenetic mechanisms such as histone modifications and DNA methylation are involved in regulating MSC differentiation [51,52]. Among the DNA methylation inhibitors, RG108 specifically inhibits the enzymatic activity of DNMTs and it has been revealed particularly useful for the experimental modulation of epigenetic gene regulation [53]. We hypothesized a reversion of the senescent state of pPDLSC after treatment with RG108 since it has been shown that RG108 caused a global decrease in DNA methylation and a local reduction of methylation levels at the promoters of OCT4 and NANOG, upregulating their RNA and protein content in bone marrow MSCs (BMSCs) [54]. OCT4, SOX-2, and NANOG are key regulators of the pluripotency state of the cells [55], and the up-regulation of DNMT1 induced by OCT4 and NANOG promotes the maintenance of the methylation status during DNA replication through the binding to its promoter, with the consequent inhibition of p16 and p21 expression and the promotion of the undifferentiated state of MSCs [56]. Our results confirmed the hypothesis, indeed we showed for the first time in pPDLSCs that RG108 upregulated stemness genes and inhibited p16 and p21 ones, thus pPDLSCs showed an answer to RG108 similar to BMSCs. The inhibition was also detected at the level of protein expression since we showed the reduction of the subset of PDLSCs co-expressing OCT4 and p21 in periodontally compromised teeth. Then, we evaluated whether this sort of “rejuvenation” of pPDLSCs after treatment with RG108 could also affect their multi-differentiation potential, showing a significantly increased adipogenic potential in pPDLSCs, while the other master genes SOX9 and RUNX2 did not change. Since in our experimental setting, pPDLSCs almost not expressing PPARγ, significantly upregulated it after RG108 treatment, we interpreted this as a positive response leading to the restoration of stemness. According to literature data, RG 108 can be used for in vivo experiments, proving safe [57], thus this molecule could potentially be a useful tool to treat periodontitis in situ, since its injection could restore and rejuvenate PDLSCs, improving their stem cell potency and protecting them from senescence.

## 5. Conclusions

The present data showed a differential phenotypic and functional behavior of hPDLSCs compared to pPDLSCs, which were characterized by a diverse adipogenic potential, and by the presence of an increased subset of cells co-expressing both markers of stemness and cellular senescence. These pPDLSCs were not in replicative senescence since they were not resistant to apoptosis. After treatment with DNMT inhibitor RG108, pPDLSCs significantly reversed the characteristics, losing the senescent phenotype. These findings shed novel light on the intrinsic properties of PDLSCs from donors with periodontitis and suggests new paths into the rejuvenation of orally-derived MSCs for autologous cell therapy applications in regenerative medicine.

## Figures and Tables

**Figure 1 biomedicines-11-02535-f001:**
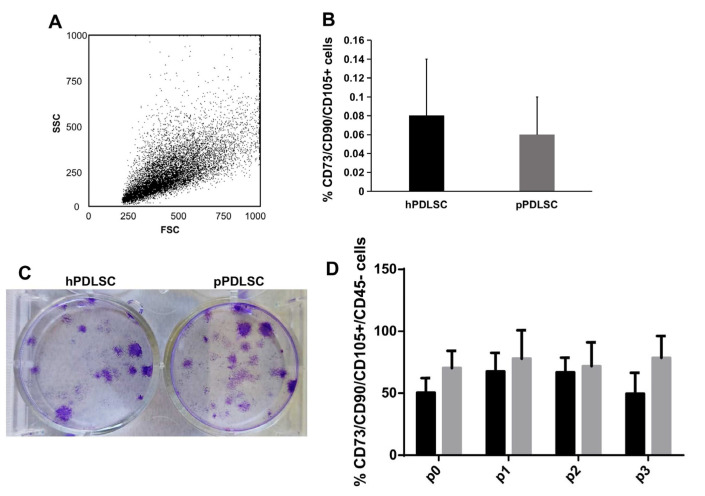
Analysis of PDLSCs after isolation. The dot plot shows the heterogeneous morphology of cells, soon after isolation from PDL (**A**), and the analysis of MSC markers showed their low expression in both hPDLSCs and pPDLSCs (**B**). Once plated both PDLSCs were able to form small colonies, stained by crystal violet (**C**). The percentage of PDLSCs co-expressing CD73, CD90, CD105, was negative for CD45 and was stable and comparable between hPDLSCs and pPDLSCs until p2, then it was significantly higher in pPDLSCs at p3, *p* < 0.05 (**D**).

**Figure 2 biomedicines-11-02535-f002:**
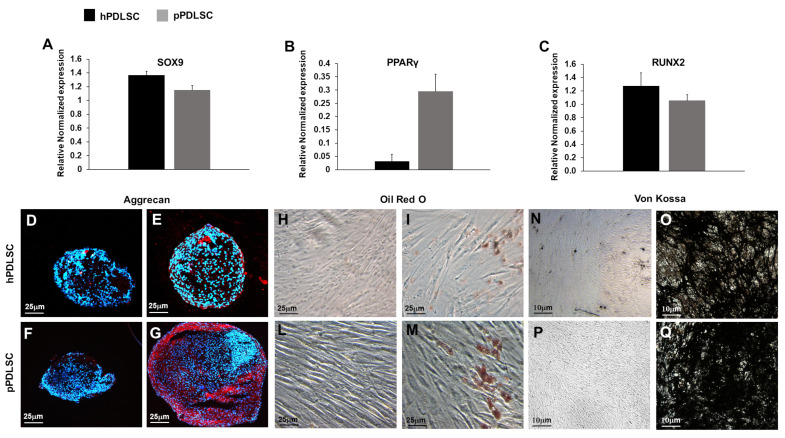
Multi-differentiating capabilities of PDLSCs. The expression of chondrocyte master gene SOX9 (**A**) and osteoblast RUNX2 (**C**) was comparable between hPDLSCs and pPDLSCs, while for, adipocytes PPARγ was higher in pPDLSCs than in hPDLSCs, *p* < 0.05 (**B**). After culture in the chondro-differentiation medium, both hPDLSC and pPDLSCs differentiated into ACAN+ chondrocytes, forming micromasses (**E**–**G**), while they did not differentiate in basal medium (**M**–**O**). Oil red O stained adipocytes (**I**–**K**) formed in the presence of adipo-differentiation medium, but not in standard medium (**H**–**J**). Osteoblasts formed mineralized nodules (**O**–**Q**) in an osteogenic medium, but not in a standard medium (**L**–**N**). Graphs indicate the mean value of gene expression ± SEM.

**Figure 3 biomedicines-11-02535-f003:**
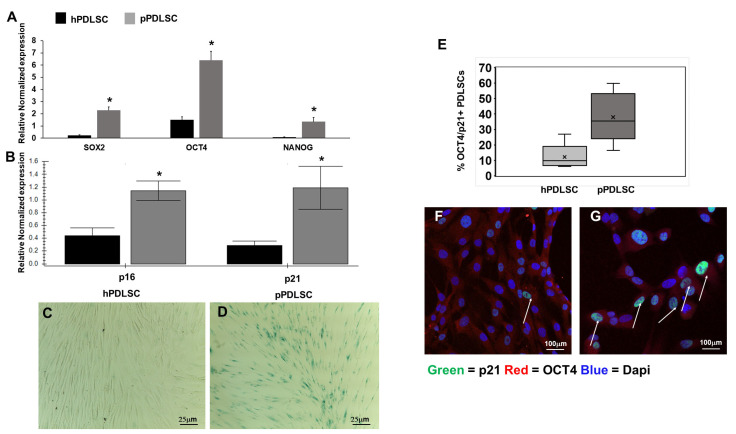
pPDLSCs contain a subset of senescent cells. The stemness genes SOX2, OCT4, and NANOG were more expressed in pPDLSCs than in hPDLSC * *p* < 0.05 (**A**), as well as the senescence genes p16 and p21, * *p* < 0.01 (**B**). Among hPDLSCs the β-Gal staining was negative, while for pPDLSCs many cells were β-Gal+, confirming the senescent state (**C**,**D**). The intra-cytoplasmic and -nucleus staining showed a subset of PDLSCs, contemporary expressing stemness (OCT4) and senescent (p21) markers (**E**), where x indicates the mean value of the % of PDLSCs positive for both OCT4 and p21. The immunofluorescence staining showed more cells co-expressing OCT4 and p21 (white arrows) in pPDLSCs than in hPDLSCs (**F**,**G**).

**Figure 4 biomedicines-11-02535-f004:**
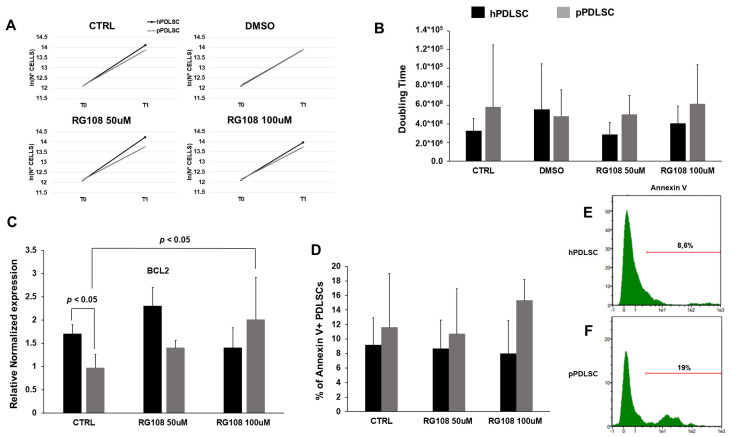
Effect of RG108 of PDLSCs. RG108 did not significantly affect growth rate (**A**) and PDT (**B**). The anti-apoptotic gene BCL2 was significantly less expressed in pPDLSCs than in hPDLSCs, *p* < 0.05 (**C**). Treatment with RG108 at 100 μM increased BCL2 expression in pPDLSCs *p* < 0.05 (**C**). A trend for a higher level of Annexin V expression was detected in pPDLSCs compared to hPDLSCs (**D**–**F**).

**Figure 5 biomedicines-11-02535-f005:**
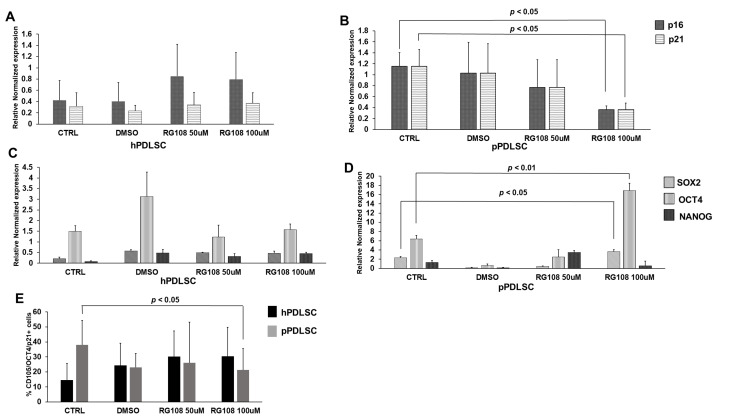
RG108 affects the senescent phenotype of pPDLSCs. RG108 did not show any significant variation of p16 and p21 expression in hPDLSCs for any of the conditions (**A**). In pPDLSCs, RG108 at 100 μM induced a significant reduction of both p16 and p21, *p* < 0.05 (**B**). The expression of the stemness genes, SOX2, OCT4, and NANOG was not significantly modulated in hPDLSCs (**C**), whereas an increase of SOX2 (*p* < 0.05) and of OCT4 (*p* < 0.01) was induced by treatment with RG108 at 100 μM (**D**). RG108 decreased the subset of pPDLSCs co-expressing OCT4 and p21 (*p* < 0.05), while no significant modulation was detected on hPDLSCs (**E**).

**Figure 6 biomedicines-11-02535-f006:**
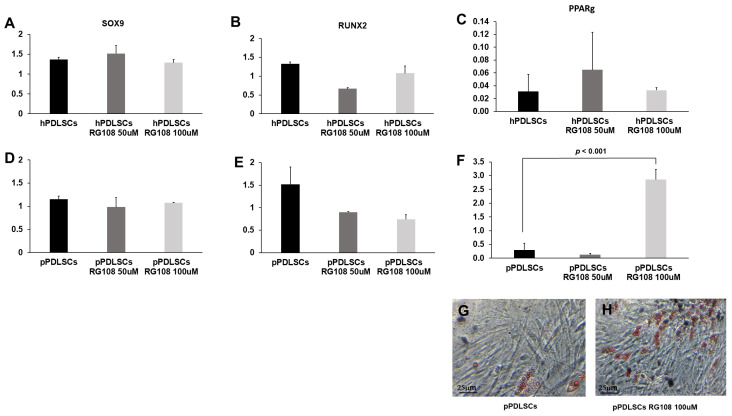
RG108 stimulates the adipogenic potential in pPDLSCs. RG108 did not affect chondrogenic and osteogenic capabilities in both hPDLSCs and pPDLSCs (**A**–**D**), while a significant up-regulation of PPARγ was detected in pPDLSCs, *p* < 0.01 (**E**,**F**). pPDLSCs treated with RG108 increased their adipogenic potential as evidenced by the major formation of cells containing lipid droplets in pPDLSCs (**G**,**H**).

**Table 1 biomedicines-11-02535-t001:** Primer sequences.

Gene	Primer	5′-3′ Sequence
PPAR*γ*	FW	AGACAACCTGCTACAAGCCC
	REV	GGGCTTGTAGCAGGTTGTCT
CDKN2A, p16	FW	AGGTCATGATGATGGGCAGC
	REV	CACCAGCGTGTCCAGGAAG
CDKN1A, p21	FW	CAAGCTCTA CCTTCCCACG
	REV	TCGACCCTGAGAGTCTCCAG
NANOG	FW	ACCCAGCTGTGTGTACTCAA
	REV	GGAAGAGTAAAGGCTGGGGT
OCT4	FW	CGAGAGGATTTTGAGGCTGC
	REV	CGAGGAGTACAGTGCAGTGA
SOX2	FW	CGAGAGGATTTTGAGGCTGC
	REV	CGAGGAGTACAGTGCAGTGA
BCL2	FW	ATGTGTGTGGAGAGCGTCAA
	REV	GGGCCGTACAGTTCCACAAA
SOX9	FW	AGCGAGCAGCAGCAGCAC
	REV	GAGTTCTGGTGGTCGGTGTAGTC
RUNX2	FW	CGAATGGCAGCACGCTATTA
	REV	TGGCTTCCATCAGCGTCAA
*β*-Actin	FW	CCCTGAAGTACCCCATCGA
	REV	AAGGTGTGGTGCCAGATTTTC

## Data Availability

The data presented in this study are available on request from the corresponding author. The data are not publicly available due to privacy.

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
