# Peer review of "Senescent Markers Expressed by Periodontal Ligament-Derived Stem Cells (PDLSCs) Harvested from Patients with Periodontitis Can Be Rejuvenated by RG108"

_biomedicines, 2023, doi:10.3390/biomedicines11092535_

Round 1

Reviewer 1 Report

Thank you for allowing me to review your article looking at different populations of PDLSCs, and the effects of RG108. Please find my questions/comments below:

Line 41-43 is not clear, are they unavailable because they’re in third molars and bicuspid teeth, or is that the only source?

Line 54-56: Explain how this was done….They took cells from different sites in the oral cavity from the same donor?

Methods 2.7 – provide a bit more information on how the staining was performed.

Results 3.1 Clarify somewhere if the stem cell population was isolated away from the contaminating cells during flow cytometry or if cells were kept as a heterogenous mix. This would be an issue, as you cannot directly attribute any subsequent measures on stem cells rather than the overall population.

Figure 2. What is the difference between e.g. H & I, N & O? Why is N & O light field but not the corresponding group for pPDLSC (P&Q)? There should be scale bars on images.

Figure 3. There should be scale bars on C,D, F & G, not just what mag. you used.

Ln 265 – take out ‘nor’.

Ln 270 – typo “hPDLSCsin”

Figure 4. – Why are the error bars so large?

Figure 5. Is RG108 used in vivo? What is the recommended/working dose? The y-axes should indicate expression of what. There is clearly a difference in the ratio of p16:p21 in the healthy group that is not seen in the perio. cells. Why?

Figure 6. Need legend so it is clear what each color bar represents. The labels on the x-axis show as incomplete, therefore not easy to follow. Why did you not confirm the different lineages the same way as in Figure 2, with microscopy images? You provided images illustrating adipogenesis, but no others.

Discussion.

Why did you only follow 3 passages? Stem cells are supposed to self-renew, and later passages may have alleviated the issue you have with mixed cell populations. It is not clear that the assays you did had the stem cell population isolated out from the heterogenous group.

Ln. 390. Did you demonstrate that they could be rescued, or are you speculating?

Some of the phrasing reads a little strangely. The grammar could be improved.

Reviewer 2 Report

Dear authors,

I consider your work very interesting and with a lot of potential, however I seem to have some points to clarify before my recommendation for publication.

A matrix question arises:

How are iPDLSCs different from pPDLSCs?

Introduction

Line 50 - that iPDLSCs display impaired immunomodulatory properties [12]. This outcome was confirmed also in a further study that compared i-PDLSCs

Lines 79 to 83 - This sentence seems to me more appropriate for the conclusions,...

Materials and Methods

Line 92 - 15% foetal bovine serum (FBS), 5% penicilin/strepto....because in all others you write in lower case (line 123 for instance).

Results

Line 200 - I do not agree with the expression "periodontitis patients" because periodontitis applies to teeth and not to individuals. There are individuals with healthy teeth and teeth with periodontitis and there are even teeth with periodontitis in which, on a certain face the inflammatory phenomenon is active, on other faces in quiescence, and may not even exist on other faces.

Line 201 - Typo "containis" and line 424 - "co-expressimg"

Lines 212 to 217 - In Figure 1 apparently the data for B and C are contradictory given that if in B we have higher values of markers for healthy cells in C the markers of these cells are always in smaller numbers compared to the teeth of individuals with periodontal disease.

Line 221 - (Fig. 2A-C),

Line 225 - (Fig. 2N-Q).

Lines 229, 342 - statistical p in italics

Figure 3 should only appear after the description in the text.

Line 277 - at 100 μM but Line 289 - 100u

Discussion

In my opinion, in the context of this work, the importance of the differences found in terms of the ability to differentiate in adipose tissue should be better discussed.

Line 317 - RG108

Line 327 - periodontal disease patients

Line 341 - PPAR-γ but Line 393 - PPARg

Line 353 - oral microbiota

Line 379 -  p16 and p21 as in line 408 for instance

Line 384 - BCL2

Round 2

Reviewer 1 Report

Thank you for editing your manuscript. I think that a couple of your responses to the reviewers should have been incorporated in the paper. For example, the differences found in ability to differentiate to adipose tissue, and the use of RG108.

Author Response

Thank you for your insights. We have now incorporated in the discussion the points the two reviewers have suggested.